# Prediction of the Topography of the Corticospinal Tract on T1-Weighted MR Images Using Deep-Learning-Based Segmentation

**DOI:** 10.3390/diagnostics13050911

**Published:** 2023-02-28

**Authors:** Laszlo Barany, Nirjhar Hore, Andreas Stadlbauer, Michael Buchfelder, Sebastian Brandner

**Affiliations:** Department of Neurosurgery, University Hospital Erlangen, Schwabachanlage 6, 91054 Erlangen, Germany

**Keywords:** deep learning, tractography, magnetic resonance imaging, corticospinal tract

## Abstract

Introduction: Tractography is an invaluable tool in the planning of tumor surgery in the vicinity of functionally eloquent areas of the brain as well as in the research of normal development or of various diseases. The aim of our study was to compare the performance of a deep-learning-based image segmentation for the prediction of the topography of white matter tracts on T1-weighted MR images to the performance of a manual segmentation. Methods: T1-weighted MR images of 190 healthy subjects from 6 different datasets were utilized in this study. Using deterministic diffusion tensor imaging, we first reconstructed the corticospinal tract on both sides. After training a segmentation model on 90 subjects of the PIOP2 dataset using the nnU-Net in a cloud-based environment with graphical processing unit (Google Colab), we evaluated its performance using 100 subjects from 6 different datasets. Results: Our algorithm created a segmentation model that predicted the topography of the corticospinal pathway on T1-weighted images in healthy subjects. The average dice score was 0.5479 (0.3513–0.7184) on the validation dataset. Conclusions: Deep-learning-based segmentation could be applicable in the future to predict the location of white matter pathways in T1-weighted scans.

## 1. Introduction

Understanding the spatial relationships between intracranial lesions and eloquent cortical and subcortical structures during neurosurgical procedures is essential in maximizing the extent of resection while improving patient safety by minimizing the risk of developing postoperative neurological deficits [1,2,3,4,5,6,7,8,9,10,11,12,13,14,15,16,17,18]. Functional magnetic resonance imaging (fMRI) [19] as well as transcranial magnetic stimulation [20] are powerful tools both in a clinical setting as well as in theoretical neuroscience, enabling non-invasive and in-vivo mapping of the human cortex. However, they provide no information about the architecture of the underlying white matter. Although MRI-based tractography currently represents the only method allowing non-invasive depiction of white matter pathways [13], it is still an evolving technique with some disadvantages from the clinical point of view such as enormous time-requirement as well as constraints with respect to accuracy and reproducibility [8,10,11,12,21,22]. 

Diffusion tensor imaging (DTI) is the most commonly used data acquisition technique in MR-based tractography in clinical practice [21] Algorithms for fiber tracking are incorporated in the software of many commercially available neuronavigation systems. Despite fiber tracking being known to be associated with the presence of false-negative fibers [8,12,21,23,24,25], numerous studies have nevertheless proven its accuracy through correlation of the results of preoperative tractography with intraoperative direct subcortical stimulation [1,2,3,5,21,23,26,27,28,29,30,31,32,33,34,35,36,37,38]. The limited reproducibility of tractography is associated with subjective elements during processing, such as different tracking parameters and stop criteria as well as the manual delineation of various regions of interests (ROI) [13,39,40].

Many studies have addressed the possible elimination of these subjective factors by proposing automation of tractography [18,22,41,42,43,44,45,46] as well as by protocoling the tracking parameters or the delineation of ROIs [18,22,34,43,47]. Others have suggested deep-learning-based methods to process the DTI sequences in order to achieve more reliable results [48] However, these approaches still mandate the time-consuming acquisition of DTI data.

Using a deep-learning-based image segmentation model can circumvent this limitation. Qi et al. recently proved the statistical superiority of this approach by comparing it to six different atlas-based automatic tractography methods [49]. The disadvantages of the atlas-based approaches such as the false-positive fibers due to the large ROIs are well known in the literature [18,22,43,47]. The aim of our study was to train a deep-learning-based image segmentation model to predict the course of the corticospinal tract on T1-weighted images and to evaluate its performance by comparing it to manual reconstruction, which represents the current clinical gold standard.

## 2. Materials and Methods

T1-weighted and pre-processed DTIs of 190 randomly selected healthy subjects from 6 different datasets were involved in this study; 90 subjects from the PIOP2 dataset of the publicly accessible Amsterdam Open MRI Collection [50] were used for the training of the image segmentation model, and 25 subjects each from the Beijing Enhanced dataset [51] and the Emotion regulation dataset [52], 15 subjects each from the Forrest Gump dataset [53] and the interTVA dataset [54], and 10 subjects each from the Test-Retest dataset [55] and the PIOP2 dataset [50] were used for the testing. The acquisition parameters of the images as well as the pre-processing are described in the original publications of each dataset. Table 1 summarizes the parameters of the datasets. Subjects with asymmetric ventricles were excluded from further processing. Since publicly available datasets were analyzed, no prior approval of the ethic commission of the Friedrich-Alexander University Erlangen–Nuremberg was required for this study.

### 2.1. Fiber Tracking and Coregistration

For each subject, we first reconstructed the corticospinal tract on both sides using the publicly available DSI Studio software (Version Chen) [56]. DTI images were aligned with the anterior commissure/posterior commissure line with a rigid transformation. The accuracy of b-table orientation was checked by automatically comparing the fiber orientations with a population-averaged template [57]. Two inclusion ROIs were drawn in every case on the directionally encoded color map. The first was placed on the two blue columns in the ventral part of the pons, and the second on the subcortical white matter of the precentral gyrus. The anisotropy threshold was 0.2, the angular threshold 45°, the step size 0.5 voxel, and the maximum length 200 mm. A total of 10,000 fibers were reconstructed on either side and a topology informed pruning performed with 16 iterations to remove false connections [58]. The results were visually controlled and fibers originating outside the precentral gyrus were manually removed using exclusion ROIs. In cases when the topology informed pruning was not applicable, false fibers were removed manually only using multiple exclusion ROIs. T1-weighted images of the same subject were imported and coregistered using an affine registration model. Coregistration was visually controlled on representative slices of the cortex, the capsula interna, the mesencephalon, as well as the pons in every case. To minimize possible inaccuracy originating from this step, coregistration parameters were adjusted in cases in which the results of the automatic registration were inaccurate. If the manual coregistration still remained unsuitable, the subject was excluded from the study. 

### 2.2. Generation of Label Images

Every voxel on the T1-weighted images containing fibers of the corticospinal tract was marked. The T1-weighted images were saved both before and after marking. Further steps were performed using the Slicer 3D v.4.11.20210226 open-source software [59,60]. To facilitate the deep learning training progress, the T1-weighed images were downsampled to a voxel size of 2 × 2 × 2 mm using a linear algorithm. A label image was created in the same dimension, and every pixel containing fibers of the corticospinal tract was marked by subtracting the unmarked T1-weighted image from the marked one.

### 2.3. Training of an Image Segmentation Model

The publicly available nnU-Net was used to train an image segmentation model to predict the course of the corticospinal tract on T1-weighted images [61]. The training process was performed in a cloud-based environment with graphical processing unit (Google Colab Pro, Google, Mountain View, CA, USA). A total of 1000 epochs were performed during the training. 

### 2.4. Performance Evaluation

Validation of the image segmentation model’s performance was carried out in a qualitative and a quantitative matter. 

Qualitative evaluation included visual inspection and correlation of the predicted with the well-known anatomical course of the corticospinal tract with special reference to the pons (through the ventral part), mesencephalon (through the middle third of the crus cerebri), capsula interna (through the posterior limb), and cortex (termination in the precentral gyrus as well as following the pattern of the minor gyri in particular cases).

Quantitative evaluation was performed by calculation of the dice score in each case. The dice score was interpreted as the number of the overlapping voxels of the corticospinal tracts in the predicted and manually segmented images multiplied by 2 and divided by the sum of the voxels containing the corticospinal tracts in both images. The widely used and accepted criteria of Landis and Koch [62] were used for interpretation. Statistical analysis was performed using the Wilcoxon test in the R programming language (version 4.2.1). Statistical significance was accepted as *p* ≤ 0.05.

## 3. Results

The course of the corticospinal tract was successfully predicted on both sides of the T1-weighted images in all subjects. The average dice score during the validation was 0.5479, with a minimum of 0.3513 and a maximum of 0.7184.

Our model performed best on the AOMIC-PIOP2 dataset with an average dice score of 0.6519 (0.4990–0.7561). The average dice score was 0.5799 (0.5037–0.6565) on the Beijing Enhanced, 0.6036 (0.4879–0.6888) on the Forrest Gump, 0.4586 (0.3513–0.6146) on the Test-Retest, 0.4733 (0.3992–0.5566) on the interTVA, and 0.5224 (0.3985–0.6563) on the Emotion regulation dataset. 

There was no statistically significant difference between the dice scores of the PIOP2 dataset and the Forrest Gump dataset (*p* = 0.11); however, this difference was significant between the PIOP 2 dataset and the Beijing Enhanced dataset (*p* = 0.01), Test-Retest dataset (*p* < 0.001), interTVA dataset (*p* < 0.001), and Emotion regulation dataset (*p* < 0.001). 

According to the scanner manufacturer, the average dice score was 0.5332 (0.3985–0.6565) on the subjects acquired with a Siemens scanner and 0.576 (0.3513–0.7561) on the subjects acquired with a Phillips scanner. There was no statistically significant difference between the performance of our algorithm on subjects acquired with a Phillips or with a Siemens scanner (*p* = 0.053).

The corticospinal tract traversed the ventral part of the pons in 100/100 cases (100%), traversed the middle-third of the crus cerebri of the mesencephalon in 100/100 cases (100%), traversed the posterior limb of the internal capsule in 100/100 cases (100%), and originated from the medial part of the precentral gyrus in 89/100 cases (89%) (Figure 1, Figure 2 and Figure 3).

The predicted pathway originated from the centrum semiovale directly under the precentral gyrus in 11/100 (11%) cases; 4 subjects (4/25, 16%) were affected in the Emotion regulation dataset, with 3 on the right side and 1 on the left side, and 7 subjects (7/15, 46.67%) were affected in the interTVA dataset, with 4 on the right side, 1 on the left side, and 2 on both sides. 

The predicted course originated from the subarachnoidal space between the minor gyri of the precentral gyrus or between the precentral and postcentral gyrus in 8/100 cases (8%) of 4 datasets, and 3 cases (3/25, 12%) of the Emotion regulation dataset, 2 cases (2/15, 13.33%) of the Forrest Gump dataset, 2 cases (2/25, 8%) of the Beijing Enhanced dataset, and a single case of the Test-Retest dataset (1/10, 10%) were affected by this error of prediction. With regard to the number of subjects with falsely segmented origins, there was no statistically significant difference between the two scanner groups (*p* = 0.06). 

False predicted areas were observed in 23/100 cases (23%) of 3 datasets, with 15 subjects (15/25, 60%) in the Emotion regulation dataset, 7 subjects (7/10, 70%) in the interTVA dataset, and 1 (1/15, 6.66%) subject in the Forrest Gump dataset being affected. According to the course of the pontocerebellar tract, the right hemisphere of the cerebellum was the most common (12/23, 52.17%) falsely segmented area. False prediction in the region parotideomasseterica was observed in 7 cases (7/23, 30.43%), in the pterygoid muscles in 3 cases (3/23, 13.04%), and in the constrictor muscles of the pharynx in 2 cases (2/23, 8.69%) (Figure 4). However, the course of the corticospinal tract was correctly predicted in 19 cases (19/23, 82.60%). In 4 cases, the predicted pathway originated from the centrum semiovale. The number of subjects with falsely segmented areas differed significantly between the groups acquired with Phillips (*n* = 1) and Siemens scanners (*n* = 22) (*p* < 0.001). 

## 4. Discussion

Tractography is the only currently available method enabling non-invasive and in-vivo visualization of white matter architecture [13] and represents an important tool in the study of normal neurobiological developmental processes [63,64] as well as of pathological changes of the white matter in various disorders such as dementia [65] or schizophrenia [66]. It has gained widespread acceptance in clinical practice as well in recent decades, especially in neurosurgery, where the most important application lies in tumor surgery. Visualization and understanding of the spatial relationship between various intracranial lesions and adjacent eloquent white matter structures, such as the corticospinal tract, the optic radiation, as well as the language-related pathways, constitute an essential component of preoperative surgical planning to minimize the risk of postoperative neurological deficits [1,2,3,4,5,6,7,8,9,10,11,12,13,14,15,16,17,18]. Moreover, the information gained can help in estimating the likelihood of a gross total resection [16] or to predict postoperative outcome [67].

Intraoperative direct subcortical stimulation is an alternate but invasive method to identify eloquent white and grey matter structures of the human brain. Its combination with preoperative tractography reduces the number of the necessary stimulations [26], leading to reduction in duration of surgery as well as the likelihood of epileptic seizures secondary to stimulation [26,67]. However, language-associated pathways as well as the visual pathway can be monitored intraoperatively with electrophysiological methods only in awake patients. Such awake surgeries may be cumbersome and time-consuming and thus cannot be performed in all patients. Moreover, in cases when awake surgery is not feasible due to patient related causes such as compliance, fatigue, or epileptic seizures, preoperative tractography remains the only tool for mapping eloquent language-related fibers during surgery. 

The deterministic DTI is a data acquisition technique in current neurosurgical practice [8,9,12,21] with several disadvantages well described in the literature: (1) the enormous time requirement, (2) the variable accuracy, and (3) the inconsistent reproducibility [8,10,11,12,21,22].

The classical steps of a neurosurgical tractography workflow are acquisition and pre-processing of the DWI data, reconstruction of fiber tracts, visual control and manual correction of the results, coregistration with anatomical images, and finally, uploading into the neuronavigation system. Fiber tracking requires manual setting of the user-defined inclusion of exclusion ROIs as well as adjustment of the reconstruction parameters. Multiple changes of the ROIs as well as readjustments of the reconstruction parameters may be necessary to achieve a satisfactory reconstruction. This trial-and-error-based approach is naturally time-consuming [41,47,68]. Moreover, manual setting of ROIs as well as interpretation of the results mandate extensive knowledge of human neuroanatomy as well as vast experience [46,47,68] Accurate tractography therefore requires a dedicated specialized neuroscientist or experienced neuroradiologist [34], which limits its availability to only a few centers [8].

The presence of existing but not reconstructed (false-negative) fibers is a well-known limitation of the deterministic DTI [8,12,21,23,24,25]. The most characteristic example is the lateral aspect of the corticospinal tract, i.e., fibers originating from the hand and face motor area [1,2,5,9,14,15,21,24,36,39,46,68,69,70,71]. This issue is associated with the inability of the DTI algorithm to resolve more than one fiber direction in a voxel, also known as the problem of the crossing fibers [11,16,26,36]. Usage of unscented Kalman filter tractography [72], algorithms based on high angular resolution diffusion imaging such as Q-ball imaging [73], or constrained spherical deconvolution tractography [9] can eliminate this limitation. However, these methods are not yet widespread in clinical practice due to the longer acquisition time of DWI scans, increased requirement of computational resources, and the additional need for specialized knowledge [8,9,12] Although it is generally accepted that these new methods produce aesthetically more realistic results, the reconstruction of non-existing (false positive) fibers is a known disadvantage. However, it should be emphasized that since it is impossible to control the rate of the false-negative and false-positive fibers in the field of tractography due to the absence of a ground truth, the accuracy of the different tractography algorithms remains open [39,43,47]. Although implementation of the consensus of an expert committee as a ground truth is a well-accepted method in neuroscience [10,12,18,74,75], it is not suitable for clinical application.

Intraoperative direct subcortical stimulation is considered the gold standard in the delineation of eloquent white matter pathways in vivo [76,77,78,79,80,81]. The positive correlation between the results of the preoperative tractography of the corticospinal tract [1,2,5,26,28,29,30,31,32,33,34,35,36,37] as well as the language-related pathways [3,23,26,27,34,38] and the intraoperative stimulation has been confirmed by many studies. Bonney et al. found the sensitivity and the specificity of the preoperative DTI in the reconstruction of the corticospinal tract to be 100% [34]. Although Bucci et al. also confirmed the positive correlation, according to their results, the probabilistic DTI delineated the corticospinal tract significantly closer to the intraoperative stimulation point than the deterministic DTI [36]. However, the presence of functionally irrelevant fibers [23], the unknown pattern of the spreading of electrical activity in the human subcortex [11,26], as well as the different stimulation parameters render precise interpretation of the results of these comparative studies difficult. 

These comparative studies suggest a reliable accuracy of deterministic DTI for clinical application, especially in the case of the corticospinal tract. Taking the abovementioned considerations into account, we chose to predict this pathway in our present study.

The variability of the intra-rater and inter-rater reproducibility is an another important disadvantage of tractography in clinical practice as well as in neuroscience research [39,40]. The noise and artefacts during acquisition [1,8,9,10,21,34,36,39,45,82,83], the individual nature of the manual placement of ROIs [8,9,10,18,34,36,39,47,83], manual adjustment of reconstruction parameters [8,10,21,34,39,83], the algorithm [8,10,21,36,39], software chosen [25,68,82], or the computer [25] used for reconstruction represent the most common causes of low reproducibility. Pathology-related factors such as peritumoral edema [8,34,36] or infiltration of the target tract by a tumor represent additive factors in a clinical context. 

To eliminate the subjective factors behind the low reproducibility, automation of the tractography workflow was proposed [18,22,41,42,43,44,45,46] Atlas-based approaches register the individual subject in a common atlas space and use its anatomical information to segment various cortical and subcortical structures as ROIs [18,22,43,47]. However, these result in relatively large ROIs due to individual anatomical variations, therefore leading to an increase in the rate of false-positive fibers [18,47]. In contrast, clustering-based methods group similar streamlines into coherent fiber tracts [42,44,46,47]. However, fibers terminating outside the main target area of the tract (such as the precommissural fibers of the fornix) could be excluded during the clustering, resulting in some false-negative fibers [18,22].

Deep-learning-based methods have been applied increasingly successfully in many studies to improve the speed, accuracy, and reproducibility of the fiber tracking process. A recent article provides a good overview of these studies [48]. Using DWI scans as input is a component of these algorithms, although the acquisition and pre-processing of the DWIs involve significant time requirements. Although this would not be an issue in the preoperative planning of elective surgeries, this additional time consumption would be unacceptable in the case of emergency surgeries. 

Using deep-learning-based image segmentation to predict the course of white matter pathways could obviate the need for acquisition of DWI scans. Qi et al. recently proved the statistical superiority of the performance of this approach by comparing it to six different atlas-based automatic tractography methods [49]. Taking the abovementioned, well-known limitations of these methods into consideration, there is a need to evaluate the performance of deep-learning-based image segmentation approaches by comparing it to the current gold standard in the clinical setting, i.e., manual segmentation. To the best of our knowledge, the present study is the first to evaluate this approach.

The main advantage of the deep-learning-based approach is that the execution of the algorithm on one image always results in the same prediction. Considering the abovementioned causes of low reproducibility, minor changes could entail significant differences in the results. 

Nevertheless, we recognize that our study has several limitations. Firstly, our model is not ready for clinical use in its present form since it predicts the course of the corticospinal tract only in healthy subjects. In the present form, the deep-learning-based image segmentation algorithm would be limited to preoperative planning only in cases without a distorted anatomy, e.g., in epilepsy surgery or functional neurosurgery, where the neuroanatomy is unaffected in most cases.

Secondly, several unavoidable factors such as acquisition errors and coregistration lead to minor inaccuracies. To minimalize inaccuracy, we used pre-processed DWI scans and manually controlled the results of the coregistration between the T1 and DWI scans in every case. Moreover, subjects without satisfying coregistration results were excluded. Furthermore, we downsampled the T1-weighted images to facilitate training speed, although using the original 1 × 1 × 1 mm voxel size could have increased the accuracy of the prediction. 

There are other possible factors that negatively affected the calculated dice score despite the qualitatively correct prediction and should therefore be considered during the interpretation of our results.

Firstly, the dimensions of the DWIs were heterogenous between different datasets as well as within a single dataset. In some cases, especially in the Emotion regulation dataset, the inferior border during the acquisition of the DWIs was defined in a horizontal plane running through the rostral part of the pons. However, our model predicted the course of the corticospinal pathway from the precentral gyrus to the medulla oblongata. 

Secondly, the limitation of the DTI technique during the fiber tracking of the corticospinal tract described in the introduction resulted in varying distances of the reconstructed pathways from the mediansagittal plane in the precentral gyrus.

Although our model performed stably on different datasets, using only one dataset acquired on only one scanner by one manufacturer during the training process is an another important limitation of our study due to possible overfitting. To address this issue, we evaluated the performance of our model on six different datasets. 

The fact that the falsely segmented areas as well as the false origination of the predicted course of the corticospinal pathway were more common in cases of datasets acquired with a Siemens scanner demonstrates the importance of the scanner type on the results. This is in line with findings of prior published literature [84].

According to the well-accepted criteria of Landis and Koch, the average dice score of our study is moderate. Taking into consideration the fact that the inter-rater voxel-based dice score was 0.62 between experts in the study of Rheault et al. [85], our deep-learning-based image segmentation model had a comparable reproducibility. 

However, more studies are required to evaluate the exact effect of the abovementioned factors, such as the influence of scanner type or acquisition parameters, and to define the most effective approach to eliminate subjective factors during the training process. Future studies should consider the use of more than one dataset acquired on scanners from different scanner manufacturers with varying resolutions during the training process. Increasing the number of subjects during the training and validation of the deep learning model’s could increase accuracy. However, a complete control over the false negative as well as false positive fibers is almost impossible due to the lack of validation of the tractography in the absence of an anatomical ground truth. However, we hypothesize that using manual segmentations performed by several experts as input for the training process could improve the accuracy as well.

## Figures and Tables

**Figure 1 diagnostics-13-00911-f001:**
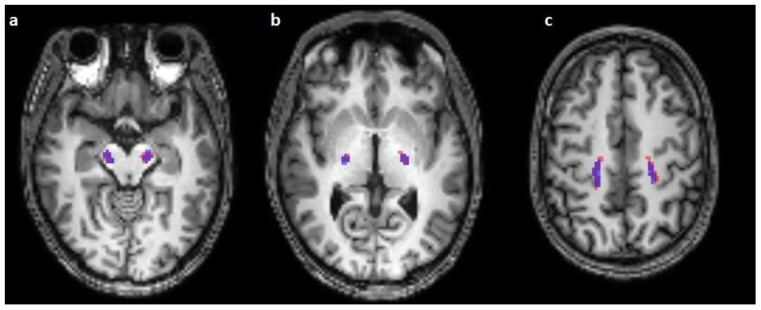
Comparison of the predicted (blue) and reconstructed (red) course of the corticospinal tract on horizontal T1-weighted images in the mesencephalon (**a**), the crus posterior of the internal capsule (**b**), and in the centrum semiovale (**c**).

**Figure 2 diagnostics-13-00911-f002:**
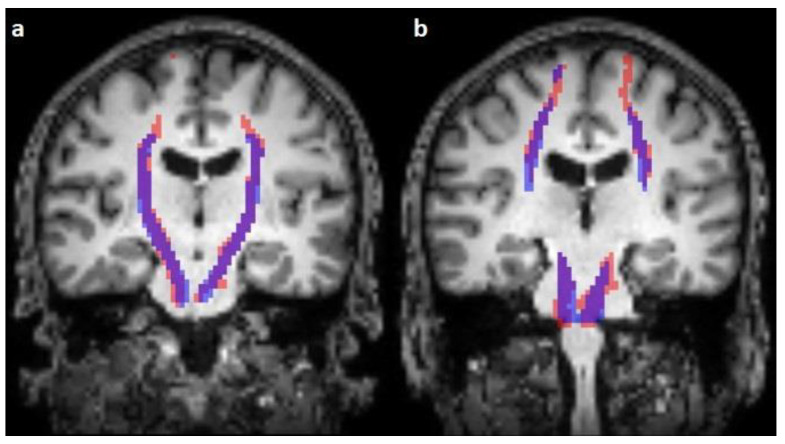
Comparison of the predicted (blue) and reconstructed (red) course of the corticospinal tract on coronal T1-weighted images (**a**,**b**).

**Figure 3 diagnostics-13-00911-f003:**
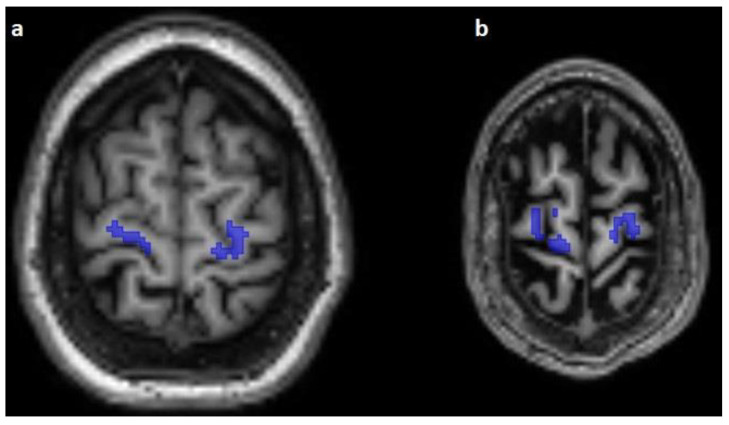
The predicted course of the corticospinal pathway (blue) in the subcortical white matter of the precentral gyrus on axial T1-weighted images. Note the absence of predicted fibers in the subarachnoidal space as well as the fibers following the pattern of the minor gyri (**a**,**b**).

**Figure 4 diagnostics-13-00911-f004:**
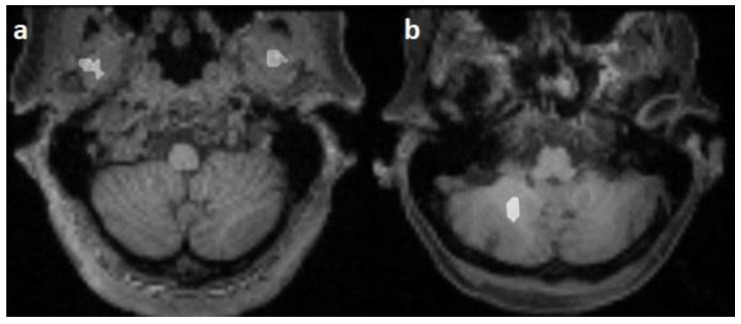
Falsely segmented areas (white) on horizontal T1-weighted images: mm. pterygoidei (**a**) and cerebellum (**b**).

**Table 1 diagnostics-13-00911-t001:** Summary of the acquisition parameters of the different datasets.

Dataset Name	Voxel Size (mm)		B-Value	Nr of DWI
	T1	DWI		
AOMIC-PIOP2	1.0 × 1.0 × 1.0	2.0 × 2.0 × 2.0	1000	32
Forrest Gump	0.7 × 0.7 × 0.7	2.0 × 2.0 × 2.0	800	32
interTVA	0.8 × 0.8 × 0.8	1.8 × 1.8 × 1.8	various	102
Emotion regulation	1.0 × 1.0 × 1.0	2.0 × 2.0 × 2.0	1000	30
Test-Retest	1.0 × 1.0 × 1.0	2.0 × 2.0 × 2.0	1000	32
Beijing	1.3 × 1.0 × 1.0	2.0 × 2.0 × 2.0	1000	64

## Data Availability

The AOMIC-PIOP2 dataset is available on https://openneuro.org/datasets/ds002790 (last accessed on 29 January 2023), the Forrest Gump dataset on https://openneuro.org/datasets/ds000113/versions/1.3.0 (last accessed on 29 January 2023), the interTVA dataset on https://openneuro.org/datasets/ds001771/versions/1.0.0 (last accessed on 29 January 2023), the Emotion regulation dataset on https://openneuro.org/datasets/ds002366/versions/1.0.0/download (last accessed on 29 January 2023), the Test-Retest dataset on https://www.nitrc.org/projects/dwi_test-retest/ (last accessed on 29 January 2023), and the Beijing enhanced dataset on http://fcon_1000.projects.nitrc.org/indi/retro/BeijingEnhanced.html (last accessed on 29 January 2023). The nnUNet is available on https://github.com/MIC-DKFZ/nnUNet (last accessed on 29 January 2023).

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
