# Peer review of "Prediction of the Topography of the Corticospinal Tract on T1-Weighted MR Images Using Deep-Learning-Based Segmentation"

_diagnostics, 2023, doi:10.3390/diagnostics13050911_

Round 1

Reviewer 1 Report

General comments:

In this manuscript, the authors describe a deep learning-based method for predicting topography of white matter tracts on MR images. This tool could potentially be useful for localizing white matter tracts on scans, which has potential application in studying nervous system development and several disease processes affecting white matter. The authors explain the study reasonably well and lay out positive results and limitations fairly. My biggest concern is the prevalence of false predicted areas (23% of cases). If I understand correctly, the main premise behind employing deep learning was to reduce subjectivity and error. How much of an improvement does this actually represent? Please address this more directly in the discussion. In addition, some additional specific explanations and discussion points would be helpful (see specific comments below).

Specific comments:

11) Abstract: talk more about the importance and applications of white matter tract visualization (development and disease, surgery)

22) Materials and Methods: please provide a more detailed explanation of the dice score in segmentation analysis

33) Discussion: the authors talk about the need for more studies in the context of improving accuracy. What approaches can future studies take, specifically to reduce the rate of false predictions?

Author Response

Thank you very much for your comprehensive review of our manuscript and the valuable observations. The revised version of our manuscripts reflects the resultant improvements as detailed below. Additionally, in accordance with your suggestion, the entire manuscript was reviewed by a native English speaker

1) How much of an improvement does this actually represent?

“The main advantage of the deep-learning based approach is that the execution of the algorithm on one image results always in the same prediction. Considering the above-mentioned causes of the low reproducibility, minor changes could entail significant differences in the results.”

2) Abstract: talk more about the importance and applications of white matter tract visualization

“Tractography is an invaluable tool in the planning of tumor surgery in the vicinity of functionally eloquent areas of the brain as well as in the research of normal development or of various diseases.”

3) Materials and Methods: please provide a more detailed explanation of the dice score in segmentation analysis

“The dice score was interpreted as the number of the overlapping voxels of the corticospinal tracts in the predicted and manually segmented images multiplied by 2 and divided by the sum of the voxels containing the corticospinal tracts in both images.”

4) Discussion: the authors talk about the need for more studies in the context of improving accuracy.

“Future studies should consider the use of more than one dataset acquired on different scanner manufacturers with varying resolutions during the training process. Increasing the number of subjects during training and validation of the deep learning model complete control over the false negative as well as false positive fibers is almost impossible due to the lack of validation of the tractography in the absence of an anatomical ground truth. However, we hypothesize that using manual segmentations performed by several experts as input for the training process could improve the accuracy as well.”

Reviewer 2 Report

A very good clinical idea (detecting corticospinal tract in a standard T1), good study and good data presentation.

Author Response

Thank you very much for your comprehensive review of our manuscript and the valuable observations. In accordance with your suggestion, the entire manuscript was reviewed by a native English speaker.